# Histological Characteristics of Follicles, Reproductive Hormones and Transcriptomic Analysis of White King Pigeon Illuminated with Red Light

**DOI:** 10.3390/ani14162320

**Published:** 2024-08-10

**Authors:** Ying Wang, Kui Zuo, Chi Zhang, Dongzhi Miao, Jing Chen, Haiming Yang, Zhiyue Wang

**Affiliations:** College of Animal Science and Technology, Yangzhou University, Yangzhou 225009, China; 13376061975@163.com (K.Z.); liyin1207@163.com (C.Z.); miaodongzhi@163.com (D.M.); cj82955@163.com (J.C.); hmyang@yzu.edu.cn (H.Y.); zywang@yzu.edu.cn (Z.W.)

**Keywords:** pigeon, follicle selection, follicular histomorphology, reproductive hormone, transcriptome

## Abstract

**Simple Summary:**

It has been observed that red light enhances egg production in pigeons. However, the underlying histological characteristics and molecular mechanisms remain unclear. In this study, fifty-four White King pigeons were selected to confirm the effect of red light on egg production in pigeons and to assess the histological characteristics of follicles, reproductive hormone levels and ovarian transcriptomics on the third day of the laying interval under both red light and white light. This study revealed a molecular basis associated with red light promoting hierarchical follicle selection, enabling a better understanding of the underlying mechanisms of red light improving egg production in pigeons.

**Abstract:**

Red light (RL) has been observed to enhance egg production in pigeons, yet the underlying histological characteristics and molecular mechanisms remain less understood. This study included fifty-four pigeons to assess follicular histology, reproductive hormones, and ovarian transcriptomics on the third day of the laying interval under RL and white light (WL). The results showed that the granulosa cell layer was significantly thicker under RL (*p* < 0.05), whereas the theca cell and connective tissue layers showed no significant differences (*p* > 0.05). Higher plasma estradiol (E_2_) levels were recorded in the RL group (*p* < 0.05), as well as follicle stimulating hormone (FSH), although progesterone (P_4_) levels were higher under WL (*p* < 0.05). Moreover, P_4_ concentrations in follicle yolk significantly decreased under RL (*p* < 0.01), with higher FSH and E_2_ levels in F1 yolk and similar increases in SF1 yolk (*p* < 0.01). Transcriptomic analysis revealed 4991 differentially expressed genes in the pigeon ovary. The protein–protein interaction network highlighted genes like *HSD11B1*, *VEGFD*, *WNT6*, *SMAD6*, and *LGR5* as potential contributors to hierarchical follicle selection under RL. This research provides new insights into the molecular basis by which RL may promote hierarchical follicle selection and improve egg production in pigeons.

## 1. Introduction

Pigeon products have become popular in the human diet for high nutrition, and they are the forth largest poultry product after chicken, duck, and goose [1,2]. However, due to their unique characteristics, with a long clutch interval, and paired pigeons only having two preovulatory follicles in a laying period [3], it is imperative to improve the egg production of pigeons. Poultry exhibits a heightened sensitivity to the red and blue portions of the light spectrum compared to humans, with longer wavelength radiation more effectively penetrating the hypothalamus, at a rate 100–1000 times greater for pigeons, than shorter wavelengths [4]. In poultry, upon photostimulation, reproductive activity is regulated by hypothalamic neuropeptides, gonadotropin-releasing hormones (GnRHs), which prompts the anterior pituitary gland to release gonadotropins, controlling the synthesis of steroid hormones and follicle development [5], which has been well studied in Japanese quail [6], songbird [7] and Dark-eyed Junco [8]. Furthermore, studies have shown that monochromatic light influences the reproductive performance of poultry. For instance, Baxter et al. (2014) [9] concluded that red light (RL) was required to stimulate the reproductive axis while green light was ineffective. Rozenboim et al. (2013) [10] found that the RL enhanced the expressions of both green and red opsin receptor. However, Gongruttananun (2011) [11] observed that RL did not enhance egg production in Thai-native hens. In our previous study, we verified that RL increased egg production in pigeons [12,13]. However, the specific mechanisms through which RL enhances pigeon egg production, particularly the histological characteristics of follicles, remain unclear.

In chickens, the ovary is a crucial organ directly linked to reproductive performance. It houses a variety of follicles of different sizes, ranging from hierarchical follicles (F1 to F6) to prehierarchical follicles, which include small yellow follicles (SYFs), large white follicles and large yellow follicles [14]. The SYF was selected and developed into hierarchical follicles daily until ovulation. Unlike chickens, paired pigeons lay just two eggs per cycle. Additionally, in our prior research, we described the morphological characteristics of pigeon follicles during the laying interval [15]. We observed that only two preovulatory follicles and small follicles were hierarchical and coupled. Furthermore, we also found that RL prompted the selection of hierarchical follicles on the 3rd day of the laying interval (LI3). Transcriptomics is commonly employed to identify genes associated with tissue function. For example, Brady et al. (2023) [16] identified ovary genes related to the ovulation of turkeys, while Yan et al. (2022) [17] focused on genes in ducks that can improve egg production. In this study, the thickness of the granulosa cell (GC) layer, theca cell (TC) layer and connective tissue (CT) layer were measured. We also assessed the reproductive hormones in the plasma and yolk of various follicles and analyzed the ovarian transcriptome of pigeons in LI3 under both RL and white light (WL). The findings offer fresh insights into the reproductive physiology of pigeons under RL and contribute to understanding the molecular mechanisms by which RL enhances egg production.

## 2. Materials and Methods

### 2.1. Animals and Sample Collection

Fifty-four paired White King pigeons (12 months old) were purchased from Taizhou Pigeon Breeding Co., Ltd. (Taizhou, China). The birds were maintained under a light regime of 15 h light and 9 h dark and water and the same food were available ad libitum. The allocation of light colors was carried out within blocks of two compartments. The pairs were housed in laying batteries (length × width × height = 40 cm × 33 cm × 30 cm). The birds were divided into two groups: one exposed to RL at 660 nm and the other to WL covering a spectrum of 400–760 nm, with each group further divided into three replicates. The light intensity was 17.50 ± 2.50 lux, which was measured with a TES-1336A light meter (TES Electrical Electronic Corp., Taiwan, China). The duration of the experiment was six months. Twelve female pigeons from each group with similar physiological states (LI3) were selected for blood sampling. From each bird, a 4 to 5 mL blood sample was drawn from the wing vein into tubes containing 10 μL of 0.8 M heparin sodium. The samples were then centrifuged at 3000× *g* at 4 °C for five min and stored at −20 °C until analysis. Concurrently, the yolks from the F1, F2 and the largest small follicle (SF1) of pigeons were used for plasma collection and were harvested to measure reproductive hormone concentrations. After gently slicing the follicles with a scalpel, the yolk was collected by penetrating the disposable syringe needle into the incision and sucking [18]. The follicles of other six female pigeons were then fixed in 4% (*v*/*v*) buffered paraformaldehyde for 24 h post-yolk removal [19]. Additionally, three female pigeons from each group in LI3 were anesthetized with isoflurane and decapitated, and their ovaries were collected, immediately frozen in liquid nitrogen, and stored at −80 °C until analysis.

### 2.2. Reproductive Hormone Concentrations Analysis

The concentrations of progesterone (P_4_), estradiol (E_2_) and follicle-stimulating hormone (FSH) were measured using enzyme-linked immunosorbent assay (ELISA) kits, following the manufacturer’s instructions, The E_2_ ELISA kit, P_4_ ELISA kit and FSH ELISA kit were purchased from ElabScience Biotechnology Co., Ltd. (Wuhan, China) [15,20]. The sensitivity of P_4_ concentration is 1.25–75 ng/mL, the sensitivity of E_2_ concentration is 12.5–700 ng/L and the sensitivity of FSH concentration is 0.1875–11.25 IU/L. The intra-assay coefficients of variation for all hormones were less than 10% and interassay coefficients of variation for all hormones were less than 15%. The ELISA Calc software v1.0 (Comple-Software, Iowa City, IA, USA) was used to calculate the concentrations of the various reproductive hormones.

### 2.3. Histological Analysis of Follicles

After fixation, the follicles were dehydrated and embedded in paraffin wax. Thin sections measuring 5 μm in thickness were prepared, stained with hematoxylin and eosin, and examined under a light microscope (Nikon, Tokyo, Japan). The thicknesses of the GC layer, TC layer and CT layer in the follicles were measured using ImageJ 2X software (National Institutes of Health, Bethesda, MD, USA). The arrangement of GC is relatively neat, and the nucleus presents a square shape; the cell nuclei of the TC and CT layer are long spindle, and the arrangement of CT cells is very loose [21].

### 2.4. Transcriptome of Pigeon Ovary

Six RNA-seq libraries were constructed from pigeon ovaries raised under red (RO1, RO2, RO3) or white (WO1, WO2, WO3) light. Total RNA was isolated from each sample using TRIzol reagent (Invitrogen, Carlsbad, CA, USA) and RNA samples were prepared following the manufacturers’ instructions. RNA integrity, purity and quantity were assessed using the 2100 Bioanalyzer (Agilent Technologies, Wilmington, DE, USA). Sequencing adaptors were attached to RNA fragments to construct the cDNA libraries, which were then sequenced on the Illumina Hiseq 4000 platform (Illumina, Inc., San Diego, CA, USA). Differentially expressed genes (DEGs) between the groups were identified using the DESeq2 R package, with significance determined by a Q-value < 0.05 and an absolute fold change greater than 2. Functional prediction and classification of the unigenes were performed using the KEGG database and GO unigene annotations. Additionally, the protein–protein interaction (PPI) network for the DEGs, encoding proteins in the pigeon ovary under RL, was analyzed using the STRING protein interaction database and visualized with Cytoscape software (version 3.9.0).

### 2.5. RT-qPCR Validation

The gene primer sequences have been listed in Table 1. Total RNA was extracted using TRIzol reagent (Invitrogen, Carlsbad, CA, USA) and reverse transcribed with the Fast Quant RT Kit (Tiangen Biotech Co., Ltd., Beijing, China). The expressions of selected genes were detected using SuperReal PreMix (Tiangen Biotech Co., Ltd., Beijing, China). Each sample was analyzed in triplicate, with GAPDH serving as an internal reference gene. Relative gene expression was calculated using the 2^−ΔΔCt^ method.

### 2.6. Statistical Analysis

Data are expressed as means ± standard deviation. The differences between groups were assessed using the independent-samples *t* test, with a significance threshold set at *p* < 0.05.

## 3. Results

### 3.1. Histological Observation of Follicles in Pigeon under RL

The thickness of the GC layer, TC layer and CT layer of follicles in pigeons under RL and WL was detected (Figure 1). In the F1 follicles, the thickness of the GC layer in RL was 15.200 μm (Table 2), which was significantly higher than in the WL group (*p* < 0.05). There were no significant differences in the thickness of the TC and CT layers between the two groups (*p* > 0.05). In the F2 follicles, the thicknesses were 13.633 μm for the GC layer, 35.667 μm for the TC layer and 49.900 μm for the CT layer. In SF1 follicles, there were no significant differences in the thicknesses of the GC, TC and CT layers between the two light conditions (*p* > 0.05).

### 3.2. Concentrations of Reproductive Hormones in Females under RL in LI3

The concentrations of E_2_, P_4_ and FSH in the plasma and yolk of females under RL in LI3 were measured. The plasma E_2_ concentration in the RL group reached 301.12 ng/L, significantly higher than in the WL group (*p* < 0.05; Figure 2A). Similarly, plasma FSH concentrations followed the same pattern, with RL promoting an increase (*p* < 0.05; Figure 2C). However, the plasma P_4_ level was significantly lower in the RL group (*p* < 0.01; Figure 2B). In the yolk, the concentration of E_2_ in F1 follicles under RL was 1.83 ng/L higher than under WL, though this difference was not statistically significant (*p* > 0.05; Figure 3A). The E_2_ levels in the SF1 yolk showed a similar pattern (*p* > 0.05), with the F2 yolk E_2_ levels under RL reaching 36.77 ng/L. The patterns of yolk P_4_ concentrations mirrored those in the plasma, being significantly higher in the RL group compared to the WL group (*p* < 0.01; Figure 3B). The FSH concentration in the F1 yolk was significantly higher in the RL group (*p* < 0.01; Figure 3C), and it was also higher in the SF1 yolk (*p* > 0.05). Overall, RL appears to positively influence certain reproductive hormone levels, which could impact egg production in pigeons.

### 3.3. Transcriptome Analysis of Pigeon Ovary under RL in LI3

Six libraries were constructed, yielding a total of 41.66 Gb of clean bases through high-throughput sequencing (Table 3). The average GC content was 49.20%, with over 93.39% of bases achieving a Q30 quality score. Between 82.07% and 87.11% of reads were successfully mapped to the pigeon reference genome (*Columba livia*). DEGs were identified in ovaries between the RL and WL groups, with a Fold-Change ≥ 2 and Q-value < 0.05. A total of 4991 genes were differentially expressed, 3769 DEGs were up-regulated and 1222 DEGs were down-regulated (Figure 4). GO analysis classified the DEGs into categories such as biological regulation, cellular processes, metabolic processes, reproduction and reproductive processes for biological processes; cell, cell part, organelle and membrane for cellular components; and binding, catalytic activity, transporter activity and molecular transducer activity for molecular functions (Figure 5). The KEGG analysis of DEGs was categorized into neuroactive ligand–receptor interaction, cell cycle, PPAP signaling pathway, steroid hormone biosynthesis, oxidative phosphorylation and the GnRH signaling pathway (Figure 6). Additionally, a PPI network was constructed, suggesting that genes such as *HSD11B1*, *VEGFD*, *WNT6*, *SMAD6* and *LGR5* may play roles in follicle selection in pigeons under RL (Figure 7).

### 3.4. Validation of Pigeon Ovary Transcriptome

Six DEGs were randomly selected for RT-qPCR to validate the transcriptome data (Figure 8). The expression patterns of these six genes in both the RL and WL groups aligned with the RNA-seq results, further substantiating the accuracy of the transcriptome analysis.

## 4. Discussion

In our previous research, the second largest follicle (F2) was not selected in LI3 [15]. However, in the current study, F2 was selected in LI3 under RL, suggesting that RL may encourage the selection of hierarchical follicles and thus enhance egg production. The thickness of the GC layer of F1 was significantly higher in RL than in WL, which was in accordance with the higher level of FSH. Yang et al. (2019) [16] concluded that a thicker GC layer aids in the transition of prehierarchical follicles to hierarchical follicles, leading to increased egg production. Similarly, the thicker GC layer of F2 under RL indicates that RL promotes the selection of hierarchical follicles and results in higher egg production.

The initial response of the GC layer to FSH is indicative of its differentiation, which facilitates the synthesis and release of steroids [22,23]. A higher FSH level in the RL group suggests earlier follicular development, and the pattern of E_2_ changes corresponds with FSH levels. Ma et al. (2020) [24] observed a similar outcome where FSH injections increased plasma E_2_ levels and expedited follicular development in hens, with higher FSH levels leading to the selection of the second hierarchical follicle in LI3 under RL. Additionally, the FSH concentrations in the follicle yolk were higher in the RL group, indicating accelerated growth of follicles. This aligns with findings by Yang et al. (2019) [19], who reported higher FSH levels in F1 compared to F2 and SYF, consistent with our results. RL stimulated the concentration of E_2_, which was significantly higher in SF1, as prehierarchical follicles are the primary source of E_2_ [15,25]. These results indicated that high concentrations of FSH and E_2_ may facilitate the selection of prehierarchical follicles to hierarchical follicles and enhance egg production under RL. However, the patterns of change in E_2_ and P_4_ levels varied under different lighting conditions. Liu et al. (2015) [26] also observed that plasma concentrations of P_4_ were higher in the WL group than in the RL group. Progesterone produced by GCs might inhibit the production of other steroid hormones [27]. Furthermore, P_4_ levels were higher in SYF compared to F1 and F2 follicles in Yangzhou geese [19], and the concentration of P_4_ in SF1 was also higher than in the F1 follicle of pigeons.

The analysis of DEGs was conducted to explore the molecular mechanisms of key genes involved in egg production under RL. The KEGG pathway analysis revealed that processes such as the cell cycle, GnRH signaling pathway and steroid hormone biosynthesis are critical for follicle selection and development. Additionally, a PPI network analysis was used to identify DEGs in core positions, including *HSD11B1*, *VEGFD*, *WNT6*, *SMAD6* and *LGR5*. Notably, 11β-hydroxysteroid dehydrogenase, which includes two enzymes, 11βHSD1 and 11βHSD2, has been identified in bovine ovaries through mRNA studies [28]. Tetsuka et al. (2010) [29] observed that the expression of *HSD11B1* mRNA increased as follicles matured in both GC and TC, indicating its importance in follicle development. Latif et al. (2005) [30] found that 11βHSD activity in follicles could be strongly inhibited by progesterone, aligning with the findings of this study where higher concentrations of P_4_ in serum and follicle yolks were noted in the WL group, correlating with lower expression of *HSD11B1*. Additionally, vascular changes are a significant aspect of follicle development. In our study, RL inhibited *VEGFD* mRNA levels; however, Kim et al. (2017) [31] suggested that *VEGFD* could stimulate early angiogenic events in the primate ovulatory follicle, highlighting that discrepancies in the results may stem from different stages of follicle development. Currently, there is limited research on the expression of *VEGFD* in relation to follicle development, indicating a need for further investigation. The WNT signaling pathway plays a crucial role in follicle growth, with *WNT6* specifically stimulating AMH, which in turn could inhibit the proliferation of GC cells [32,33]. WL significantly stimulated the expression of WNT6 in pigeon ovaries, which may explain the delayed selection of follicles observed under WL conditions. This finding aligns with the known role of the WNT signaling pathway in follicle development. Additionally, *LGR5*, identified as a potential target of WNT signaling, encodes a G-protein-coupled receptor [34]. This receptor is related to the hormone receptors for *FSH* and *LH*, which are critical in regulating reproductive processes. The interaction between WNT signaling and *LGR5* suggests a complex network influencing follicle development and hormone interaction [35]. Rastetter et al. (2014) [36] demonstrated that *LGR5*-positive cells can develop into cortical adult GC. In our study, we observed that RL increased *LGR5* mRNA levels, which may accelerate the differentiation of GCs and promote the selection of hierarchical follicles. The SMAD family is divided into receptor-activated, common-partner, or inhibitory SMADs, with *SMAD6* functioning as an inhibitor of *BMP* signaling [37]. Several BMPs are known to act as autocrine/paracrine regulators of ovarian follicle development [38]. Domingues et al. (2023) [39] demonstrated that *SMAD6* could significantly inhibit GC proliferation and the rate of follicle growth. Consistently, in our study, *SMAD6* was significantly up-regulated in the WL group, which suppressed the proliferation of GCs and delayed the selection of hierarchical follicles. Our results provide a mechanistic insight into how light spectrum can influence follicular dynamics through molecular pathways.

## 5. Conclusions

In conclusion, our study is the first to present the histological characteristics of pigeon follicles, along with the reproductive hormone levels in plasma and yolk and an ovary transcriptome analysis under RL. RL was found to stimulate the proliferation of GC and TC layers, as well as enhance the secretion of FSH and E_2_ in both plasma and yolk. This stimulation promotes the selection of hierarchical follicles and increases egg production in pigeons. Additionally, an ovary transcriptome analysis was performed, where 3769 DEGs were up-regulated and 1222 DEGs were down-regulated; furthermore, key genes (*HSD11B1*, *VEGFD*, *WNT6*, *SMAD6*, and *LGR5*) involved in follicle selection were identified, providing insights into the molecular mechanisms through which RL improves egg production in pigeons.

## Figures and Tables

**Figure 1 animals-14-02320-f001:**
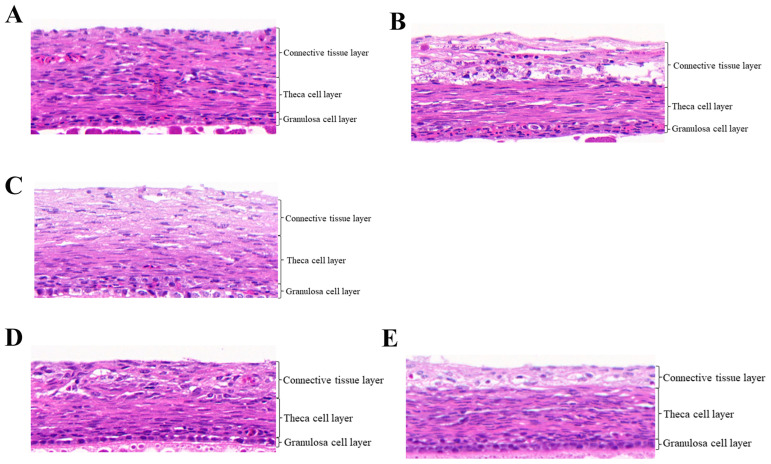
Histological characteristics of pigeon follicles under red light. (**A**) F1 follicle under RL. (**B**) F1 follicle under WL. (**C**) F2 follicle under RL. (**D**) SF1 follicle under RL. (**E**) SF1 under WL.

**Figure 2 animals-14-02320-f002:**
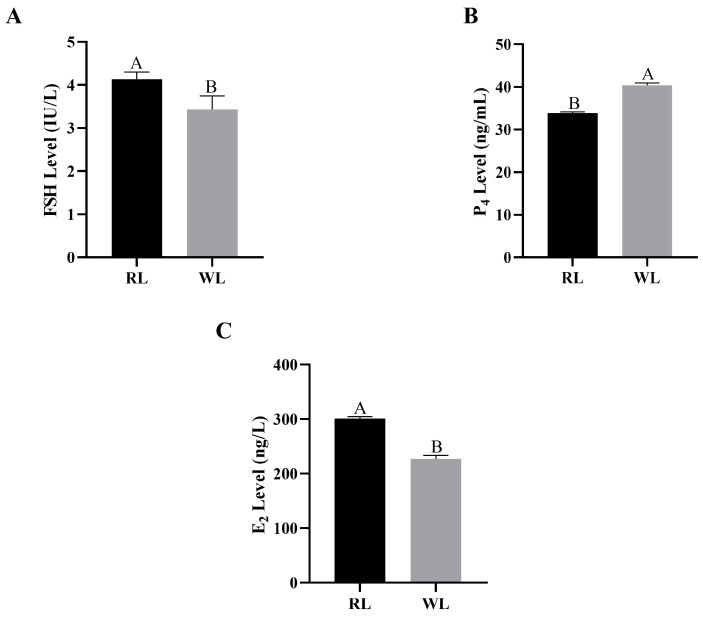
The concentration of estradiol (E_2_, df = 16, **A**), progesterone (P_4_, df = 16, **B**) and follicle-stimulating hormone (FSH, df = 9, **C**) in pigeon plasma under red light. Values marked with different capital letters on the bars are extremely significantly different (*p* < 0.01).

**Figure 3 animals-14-02320-f003:**
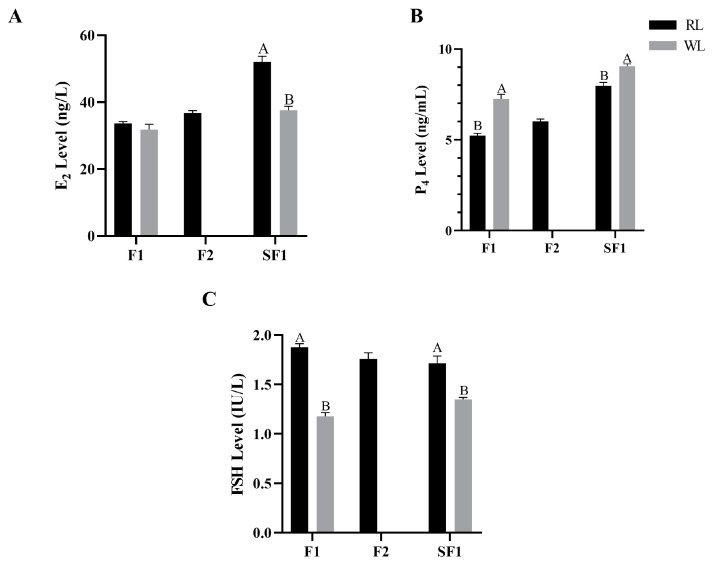
The concentration of estradiol (E_2_, df = 4, **A**), progesterone (P_4_, df = 4, **B**) and follicle-stimulating hormone (FSH, df = 4, **C**) in pigeon follicle yolk under red light. Values marked with different capital letters on the bars are extremely significantly different (*p* < 0.01).

**Figure 4 animals-14-02320-f004:**
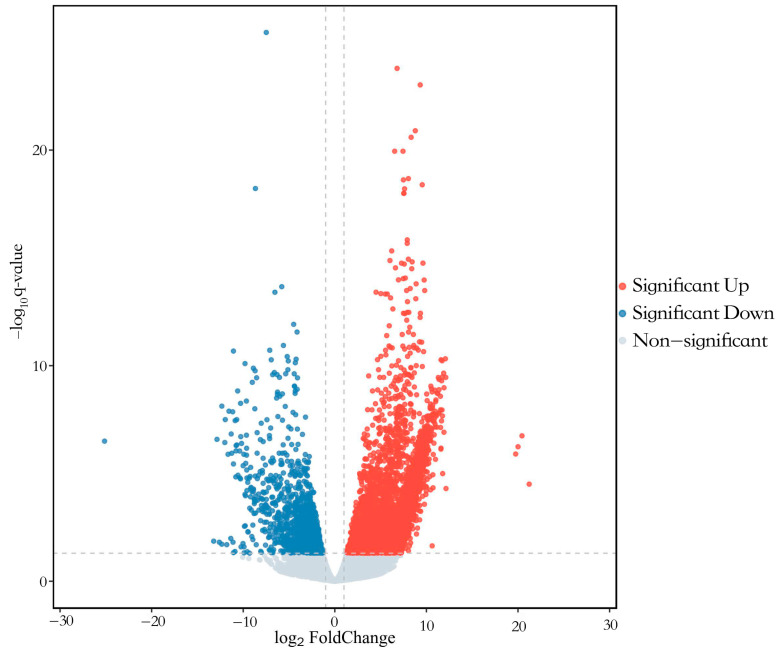
The volcano plot of differentially expressed genes.

**Figure 5 animals-14-02320-f005:**
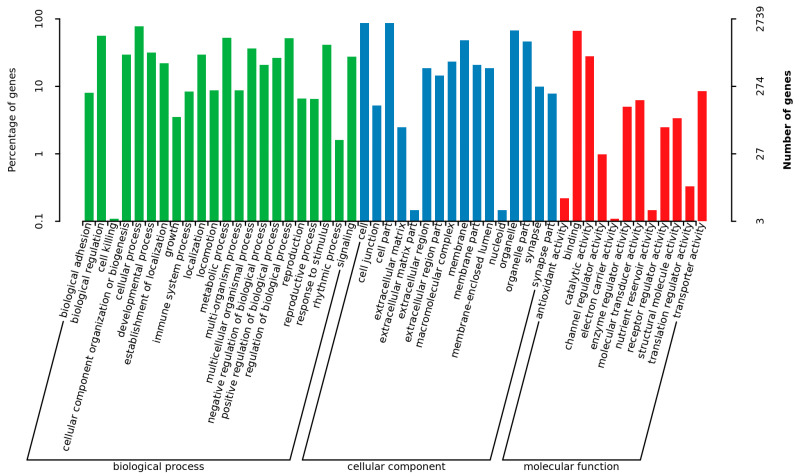
GO enrichment analysis of differentially expressed genes.

**Figure 6 animals-14-02320-f006:**
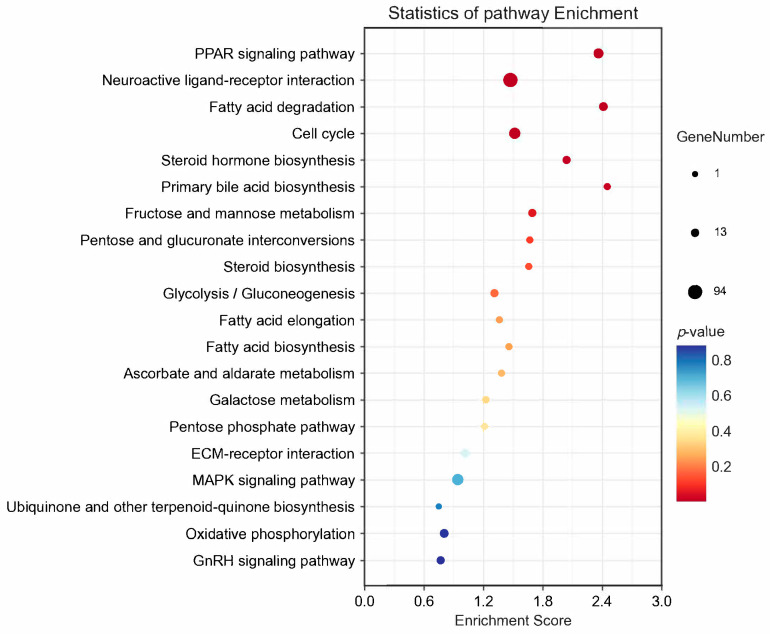
KEGG enrichment analysis of differentially expressed genes.

**Figure 7 animals-14-02320-f007:**
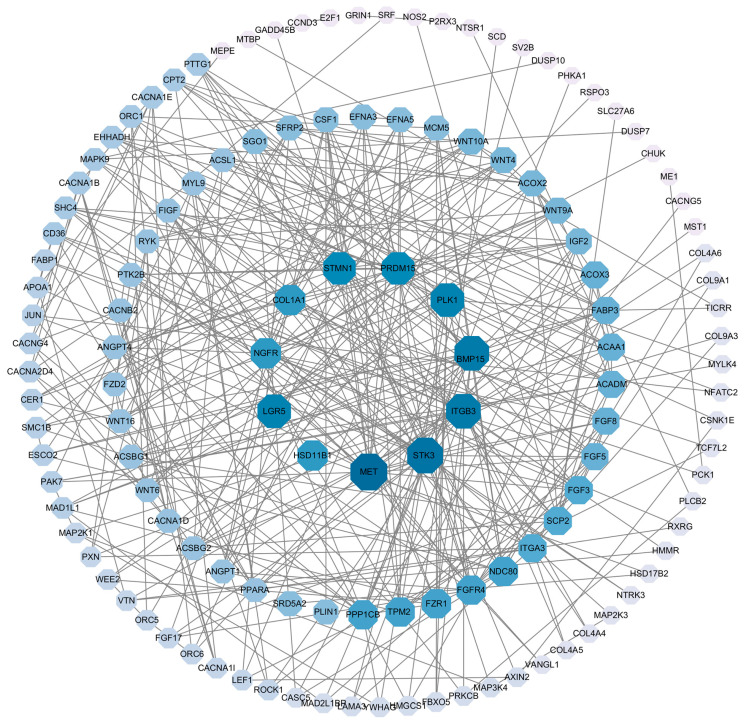
Protein–protein interaction (PPI) network of differentially expressed genes.

**Figure 8 animals-14-02320-f008:**
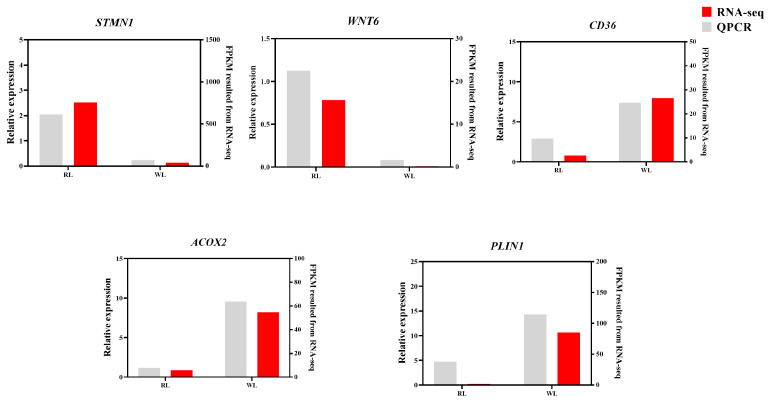
The validation of RNA-Seq using RT-qPCR.

**Table 1 animals-14-02320-t001:** Primers for RT-qPCR.

Gene	Primer Sequence (5′→3′)	Length
*STMN1*	F: ATGCTGAATATCTGTTACACGTC R: CCATTTTGTTCCGCGTGTC	130 bp
*WNT6*	F: CCAGCAGTTCATGGATGCCAA R: AAACGTCCGGCTTCATTGTTG	169 bp
*CD36*	F: CCCAAAGAAAATATCACGGAA R: ATATCAGGTTCAAAACGAGCAA	80 bp
*AOCX2*	F: AAGTGAACGCCACACGTCT R: CGGTTACCACTCAGCATCGCTTG	169 bp
*ANGPT4*	F: TCTACACCCTGCACATCACC R: TCCATGTCGCAGTACGCCTT	58 bp
*CACNB2*	F: GACGCTGATACCATTAACCAC R: TACATCAAACATTTCGGGAGG	171 bp
*GAPDH*	F: CTCTACTCATGGCCACTTCCG R: ACAACGTATTCAGCACCAGC	138 bp

**Table 2 animals-14-02320-t002:** The thickness of the follicular membrane in pigeon follicles in the RL and WL group.

Group ^1^	Granulosa Cell Layer (μm)	Theca Cell Layer (μm)	Connective Tissue Layer (μm)
RLF1 ^2^	15.200 ± 1.210 ^a^	41.866 ± 2.571	35.664 ± 4.276
WLF1	8.767 ± 0.736 ^b^	38.433 ± 2.444	49.267 ± 6.631
RLF2	13.633 ± 0.884	35.667 ± 2.267	49.900 ± 1.513
RLSF1	7.100 ± 0.451	37.267 ± 4.037	32.500 ± 3.843
WLSF1	7.133 ± 0.939	30.700 ± 3.009	40.100 ± 1.815

^1^ Values marked with different small letters on the bars are significantly different (*p* < 0.05). ^2^ Abbreviations: RLF1 = the largest follicle in the red light group; WLF1 = the largest follicle in the white light group; the df of follicular membrane of RLF1 vs. WLF1 is 4; RLF2 = the second largest follicle in the red light group; RLSF1 = the largest small follicle in the red light group; WLSF1 = the largest small follicle in the white light group; the df of the follicular membrane of RLSF1 vs. WLSF1 is 4.

**Table 3 animals-14-02320-t003:** Summary of the sequences’ assembly from pigeon ovaries under red light.

Samples ^1^	Raw Read/M	Clean Reads/M	Clean Bases/G	Mapped Reads/%	Q30/%	GC/%
RO3-1	48.95	47.01	6.87	83.33	94.45	49.92
RO3-2	50.38	47.88	6.94	83.92	94.36	48.95
RO3-3	48.20	47.22	6.95	87.11	96.04	48.85
WO3-1	51.24	48.39	6.99	84.26	93.39	49.31
WO3-2	52.81	49.15	7.01	82.07	94.07	49.16
WO3-3	48.66	47.23	6.90	86.57	96.34	49.00

^1^ Abbreviations: RO3 = RNA-seq library constructed from pigeon ovaries raised under red light; WO3 = RNA-seq library constructed from pigeon ovaries raised under white light.

## Data Availability

The data were not deposited in an official repository; the accession number can be found below: NCBI; PRJNA1125931. Data are available upon request to the corresponding author.

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
