# Peer review of "Histological Characteristics of Follicles, Reproductive Hormones and Transcriptomic Analysis of White King Pigeon Illuminated with Red Light"

_animals, 2024, doi:10.3390/ani14162320_

Round 1

Reviewer 1 Report

Comments and Suggestions for Authors

The manuscript is short and concise, but it touches on the most important issues. The interdisciplinary nature of the analysis performed deserves praise.  I am convinced that it is worthwhile for the authors to continue this research by expanding it with further aspects. The following is my list of comments:

Introduction: There is a lack of a clearly described research hypothesis. I am missing information on why to increase the laying rate of pigeons and the purpose of the research. Only the last sentence of the introduction refers to it.

I would also add how birds perceive light and how this affects their reproduction. In addition to receptors in the retina, receptors are located directly in the brain. It is impossible to discuss the effect of light in isolation from the nervous system that controls seasonal reproduction. The growth of follicles and the expression of genes in them will result from this regulation.

Animals and Sample Collection: Under what conditions were the pigeons kept? In pairs? In cages? An aviary? What cage dimensions? But more important: what does it mean that the groups were divided into 3 subgroups? What are the subgroups indeed? Yolks from the F1, F2 were collected from the same birds blood samples were taken? How those yolks were collected? From description, I understand only 3 females were decapitated and ovaries collected. Please clarify how each material/sample was collected.

Histological Analysis of Follicles: How many samples/photos were finally compared?

Statistics: differences between groups were compared by T-test, and what was compared by ANOVA?

There should be mentioned/explained why there is no data for group FW WL.

Reviewer 2 Report

Comments and Suggestions for Authors

The article has been a special study that provides important information. It will make important contributions to the literature. It can be published after the necessary edits are made.

Review report;

Title: Histological characteristics of follicles, reproductive hormones and transcriptomic analysis of White King pigeon illuminated with red light

The study revealed the effects of red light on reproductive hormone levels, follicle histology and ovulation interval in pigeons..

Title: The title is compatible with the content.

Simple Summary: The findings are written in a reflective and understandable manner, appropriate to the content of the article.

Abstract: The findings are written in a reflective and understandable manner, appropriate to the content of the article.

Keywords: It is written in accordance with the content of the study and is understandable.

Introduction: The introduction section is simple, understandable and explains the purpose of the study. For this reason, I found it very nice.

Materials and Methods:

Brief information should be given about the content of the feed consumed. It should be noted that both groups consumed the same feed and at the same level.

Was it raised in a cage system?

Number of animals per square meter?

Providing the above information will be important resources for other studies.

Statistical Analysis is suitable.

We should write the literature sources of the methods. If there is. If you created it, you should write it down.

Results: Pictures and figures should be more visible.

Findings appropriate to the purpose of the study are given.

Discussion: The discussion section is given in accordance with the study findings. Necessary explanations have been made. It was found appropriate. Reference is made to the literature.

Conclusions: It is descriptive of the work and informative to the reader. However, important information can also be presented from the RESULTS.

Owerall:

The article has been a special study that provides important information. It will make important contributions to the literature. It can be published after the necessary edits are made.

Reviewer 3 Report

Comments and Suggestions for Authors

Manuscript ID: animals-3084150

Histological characteristics of follicles, reproductive hormones and transcriptomic analysis of White King pigeon illuminated with red light

Wang et al attempted to show the effect of red light on the development of follicles, egg production and reproductive hormones. Moreover, authors performed histological characteristics of pigeon along with the reproductive hormone levels in plasma and yolk and transcriptome analysis of ovary to identify the genes expressed during the red light. Authors found that Red light stimulate the proliferation of GC and TC layers as well as enhance the secretion of FSH and E2 in both plasma and yolk. This stimulation promotes the selection of hierarchical follicles and increases egg production in pigeons. The manuscript is nicely design and conceived and well written. I indeed enjoyed reading the paper. I have some minor points for revision in the manuscript (see my comments below and – for the ease of the authors).

Line 78. Please provide the complete methodology for the preparation of yolk samples for reproductive hormone analysis

Line 146. According to the methodology provided for blood sample preparation “blood sample was drawn from the 75 wing vein into tubes containing 10 μL of 0.8 M heparin sodium. The samples were then 76 centrifuged at 3,000 × g at 4 for five min and stored at -20 until analysis” It should be called plasma not serum. Please replace the term serum from plasma throughout the manuscript.

Line 146. Hormones are presented in IU/g. Essentially, IU/g used for the measuring the vitamins. Hormones are usually presented in picogram or nanogram per millilitre of blood plasma. Please change throughout the manuscript.

Line 203. Remove “study”

Line 206. Concluded or found that. use only one.
